# Variations in the Mood States during the Different Phases of COVID-19’s Lockdown in Young Athletes

**DOI:** 10.3390/ijerph18179326

**Published:** 2021-09-03

**Authors:** Abraham Batalla-Gavalda, Pau Cecilia-Gallego, Fernando Revillas-Ortega, Jose Vicente Beltran-Garrido

**Affiliations:** 1EUSES Escola Universitària de la Salut i l’Esport, Rovira i Virgili University, 43870 Tarragona, Spain; a.batalla@euseste.es (A.B.-G.); pau.cecilia@euseste.es (P.C.-G.); 2Centre de Tecnificació Terres de l’Ebre, 43870 Tarragona, Spain; frevillas@psicosport.org; 3Psicosport—Serveis de Psicologia de l’Esport, 43500 Tortosa, Spain

**Keywords:** COVID-19, quarantine, POMS

## Abstract

The pandemic caused by SARS-CoV-2 led to a situation of home lockdown. Competitions, training, and sports activities were canceled for much of this period. This situation could have affected the physical and mental health and the mood states of young athletes. The aim of the study was to evaluate the variations in the mood states of young athletes of a highly competitive level during the different phases of lockdown in. The mood states of 45 Spanish youth athletes confined during the 10-week lockdown due to the COVID-19 pandemic -19 were assessed, using the Spanish-translated version of the POMS questionnaire, at the start of the home lockdown and at the start and at the end of the partial lockdown. The Total Mood Disturbance scores at week 10 were lower than those obtained in week 1 (*p_Bonferroni_* = 0.031). The depression scores at week 6 (*p_Bonferroni_* = 0.048) and at week 10 (*p_Bonferroni_* < 0.001) were lower than those obtained in week 1. The confusion scores at week 10 were lower than those obtained in week 1 (*p_Bonferroni_* = 0.002). These variations differed between team and individual sports. In conclusion, the lockdown produced changes in the young athletes’ mood states that should be considered when trying to optimize their physical and mental performance.

## 1. Introduction

Due to the home lockdown and subsequent de-escalation decreed by the Government of Spain on 15 March 2020 as a result of the pandemic situation caused by SARS-CoV-2 and COVID-19 (Spain, 2020), sports competitions, training and sports activities were canceled for a great part of that period. This situation with no training and competition could have affected the physical health, mental health and mood states of athletes [1] and could, in some cases, have generated moderate and high levels of stress [2], depression [3] and anxiety [4]. Studies such as that of Leguizamo et al. [2] point out that, in terms of emotional distress, a situation of lockdown could be compared to a period of injury.

The international literature has shown that both mental health and mood states are affected by the levels of the physical activity performed. Low levels of physical activity are associated with a greater increase in anxiety [5] and stress [6]. In contrast, high levels of physical activity improve mood states and subjective well-being [7], help establish self-esteem in children and adolescents [3] and decrease the intensity of stress, anxiety and depression [8].

Studies prior to lockdown, with a general population sample, showed that high-frequency physical activity (four times per week or more) was associated with improved mood state and significantly decreased levels of stress, distrust and anger [9]. In addition, Chekroud et al. [10], with a sample of more than 1.2 million American adults, showed a negative association between a higher frequency of exercise and levels of stress, depression and emotional problems. Furthermore, studies conducted during lockdown show that as physical activity levels decrease, levels of negative states such as tension, depression and anger increase [3,11].

Mon-López et al. [12] demonstrated that there is a relationship between mood states and the amount of sleep in professional and non-professional soccer players during lockdown. However, the quality of sleep showed values in accordance with the existing literature, due to the fact that these athletes have extensive experience and high self-control. In another study, Mon-López et al. [13] reported an increase in stress and anxiety levels in professional handball players. However, the study by Leguizamo et al. [2], with a sample of 310 professional athletes from different cultures and disciplines, showed that despite the pandemic, experience with and the use of coping strategies allowed these athletes to maintain an optimal mood state profile for sports performance, even during lockdown.

As we described, studies on athletes focused on the performance of adult populations. However, we have no information regarding how lockdown or restrictions of movement affect the mood states of young competitive athletes in Spain. It is hence hypothesized that the period of lockdown negatively affected the mood states of young Spanish elite athletes. Therefore, the aim of the present study was to evaluate the variations in the profile of mood states during the different phases of lockdown (severe lockdown and de-escalation) in young athletes participating in high-level competitions.

## 2. Materials and Methods

### 2.1. Experimental Approach to the Problem

The mood states of Spanish youth athletes confined during the 10-week lockdown due to the COVID-19 pandemic -19 were assessed at week 1 (the start of the home lockdown), week 6 (the end of the home lockdown and start of the partial lockdown) and week 10 (the end of the partial lockdown).

#### Participants

A convenience sample of 80 youth athletes from a sports technicization center were selected to participate in the study. Only the participants who completed the questionnaire at the different time points were included in the study. Finally, 45 athletes from different sport disciplines (athletics, handball and swimming) were included. The sample characteristics are shown in Table 1.

The study was conducted according to the guidelines of the Declaration of Helsinki and approved by the Institutional Review Board of Consell Català de l’Esport (protocol code 0981/7736). Informed consent and assent were obtained from all subjects and their parents when the participants were minors.

### 2.2. Procedures

Mood states were assessed using the Spanish translated version of the POMS questionnaire [14]. This is a 58-item inventory of six subscales: tension–anxiety, depression–dejection, anger–hostility, vigor–activity, fatigue–inertia, and confusion–bewilderment. Each item was responded to by the participants on a 5-point Likert scale anchored by 0 (not at all) and 4 (extremely). The score of each factor is obtained by the sum of the answers of the items that compose each subscale. All items have the same direction except two: “relaxed” from the tension–anxiety subscale and “efficient” from the confusion–bewilderment subscale, which are inverted. Total Mood Disturbance (TMD) is obtained with the sum of the subscale scores. It should be noted that the vigor–activity subscale is subtracted instead of summed. In order to avoid negative scores, a 100-point constant was added to the total score.

### 2.3. Statistical Analyses

The assumption of normality was assessed using the Shapiro–Wilk test and by exploring the Q–Q plots and histograms of residuals. To compare the mood state scores of the different time points (week 1, 6 and 10) a Friedman’s ANOVA was used. When significant within-time points effects were reported, post hoc comparisons were performed with the Connover correction.

To compare the mood state scores of the athletics, handball and swimming groups and the mood state scores between sex groups, a Kruskal–Wallis ANOVA was used. When significant between-group effects were reported, Dunn’s post hoc comparisons were performed with the Bonferroni correction.

All statistical analyses were performed using JASP (version 0.14.1; JASP Team (2020), University of Amsterdam, Amsterdam, the Netherlands). The level of significance was set at 0.05 for all tests.

## 3. Results

The results of the mood state scores at different time points are shown in Table 2.

The results of the mood state scores of both sex at the different time points are shown in Appendix A.

The results of the mood state scores of the three sports at the different time points are shown in Appendix A.

The POMS profile of the three time points is shown in Figure 1.

### 3.1. Total Mood Disturbance

Data revealed a significant main effect of time (*p* = 0.031, W = 0.81). Post hoc tests showed a significant difference between week 1 and week 10 scores (*p_Bonferroni_* = 0.031); the TMD scores at week 10 were lower than those obtained in week 1.

At week 1, between-sports analyses revealed a significant main effect of sport (*p* = 0.050). Post hoc tests showed a significant difference between handball and athletics scores (*p_Bonferroni_* = 0.024); the TMD scores of the handball players were higher than those obtained by the athletes.

At week 6, between-sports analyses revealed a significant main effect of sport (*p* = 0.034). Post hoc tests showed a significant difference between handball and athletics scores (*p_Bonferroni_* = 0.016); the TMD scores of the handball players were higher than those obtained by the athletes.

At week 10, between-sports analyses revealed a significant main effect of sport (*p* = 0.032). Post hoc tests showed a significant difference between handball and athletics scores (*p_Bonferroni_* = 0.013); the TMD scores of the handball players were higher than those obtained by the athletes.

A non-significant main effect of sex was shown at week 1 (*p* = 0.186), at week 6 (*p* = 0.704) and at week 10 (*p* = 0.718).

### 3.2. Tension

Data revealed a non-significant main effect of time (*p* = 0.070, W = 0.76).

Neither at week 1 (*p* = 0.386) nor at week 10 (*p* = 0.439) did between-sports analyses reveal a non-significant main effect of sport.

At week 6, between-sports analyses revealed a significant main effect of sport (*p* = 0.043). Post hoc tests showed a significant difference between handball and swimming scores (*p_Bonferroni_* = 0.020); the tension scores of the handball players were higher than those obtained by the swimmers.

A non-significant main effect of sex was shown at week 1 (*p* = 0.640), at week 6 (*p* = 0.771) and at week 10 (*p* = 0.505).

### 3.3. Depression

Data revealed a significant main effect of time (*p* < 0.001, *W* = 0.70). Post hoc tests showed significant differences between week 1 and week 6 scores (*p_Bonferroni_* = 0.048); the depression scores at week 6 were lower than those obtained in week 1. Additionally, significant differences between week 1 and week 10 scores were found (*p_Bonferroni_* < 0.001); the depression scores at week 10 were lower than those obtained in week 1.

At week 1, between-sports analyses revealed a non-significant main effect of sport (*p* = 0.339).

At week 6, between-sports analyses revealed a significant main effect of sport (*p* = 0.028). Post hoc tests showed a significant difference between handball and athletics scores (*p_Bonferroni_* = 0.012); the depression scores of the handball players were higher than those obtained by the athletes.

At week 10, between-sports analyses revealed a significant main effect of sport (*p* = 0.027). Post hoc tests showed a significant difference between handball and athletics scores (*p_Bonferroni_* = 0.012); the depression scores of the handball players were higher than those obtained by the athletes.

A non-significant main effect of sex was shown at week 1 (*p* = 0.528), at week 6 (*p* = 0.950) and at week 10 (*p* = 0.588).

### 3.4. Anger

The results showed a non-significant main effect of time (*p* = 0.166, *W* = 0.65).

Neither at week 1 nor at week 10 did between-sports analyses reveal a non-significant main effect of sport (*p* = 0.392 and *p* = 0.067, respectively).

At week 6, between-sports analyses revealed a significant main effect of sport (*p* = 0.016). Post hoc tests showed a significant difference between handball and athletics scores (*p_Bonferroni_* = 0.005), and between handball and swimming scores (*p_Bonferroni_* = 0.030); the anger scores of the handball players were higher than those obtained by the athletes and the swimmers.

A non-significant main effect of sex was shown at week 1 (*p* = 0.840), at week 6 (*p* = 0.570) and at week 10 (*p* = 0.353).

### 3.5. Vigor

The results showed a non-significant main effect of time (*p* = 0.443, W = 0.79).

Between-sports analyses revealed a non-significant main effect of sport at week 1 (*p* = 0.434), at week 6 (*p* = 0.294) and at week 10 (*p* = 0.253).

A non-significant main effect of sex was shown at week 1 (*p* = 0.535), at week 6 (*p* = 0.960) and at week 10 (*p* = 0.783).

### 3.6. Fatigue

The results showed a non-significant main effect of time (*p* = 0.245, *W* = 0.73).

Between-sports analyses revealed a non-significant main effect of sport at week 1 (*p* = 0.138), at week 6 (*p* = 0.091) and at week 10 (*p* = 0.307).

A non-significant main effect of sex was shown at week 1 (*p* = 0.304), at week 6 (*p* = 0.425) and at week 10 (*p* = 0.580).

### 3.7. Confusion

Data revealed a significant main effect of time (*p* = 0.001, *W* = 0.75). Post hoc tests showed significant differences between week 1 and week 10 scores (*p_Bonferroni_* = 0.002); the confusion scores at week 10 were lower than those obtained in week 1.

At week 1, between-sports analyses revealed a non-significant main effect of sport (*p* = 0.053).

At week 6, between-sports analyses revealed a significant main effect of sport (*p* = 0.004). Post hoc tests showed a significant difference between handball and athletics scores (*p_Bonferroni_* = 0.001); the confusion scores of the handball players were higher than those obtained by the athletes.

At week 10, between-sports analyses revealed a significant main effect of sport (*p* = 0.021). Post hoc tests showed a significant difference between handball and athletics scores (*p_Bonferroni_* = 0.001); the confusion scores of the handball players were higher than those obtained by the athletes.

A non-significant main effect of sex was shown at week 1 (*p* = 0.282), at week 6 (*p* = 0.595) and at week 10 (*p* = 0.827).

## 4. Discussion

The present study describes the impact of lockdown and de-escalation, from March to May 2020 in Spain, on the mood states of a group of young athletes at a highly competitive level. The most important findings of this study were the decrease in the values of depression, confusion and TMD at the end of lockdown and de-escalation. Moreover, these variations were different between athletes of team sports and those of individual sports, but were not relevant between sexes.

The studies carried out during the pandemic show a decrease in the physical activity in the general population and sports-specific training in athletes when compared to the levels of the periods prior to lockdown [12,15,16,17,18]. This is related to an increase in negative mood states such as depression, tension [11], stress and anxiety [4,5]. Previous research shows that people who worked out constantly before the outbreak of the pandemic showed a greater decrease in mood states [19] and a greater loss of sleep quality [11,13,16] than people who practiced less physical activity before lockdown.

Additionally, the environment of athletes may affect mood states [20]. In fact, O’Connor et al. [21] pointed out that the negative effects increase in situations of training restriction, in periods of isolation and if competitions are canceled. Additionally, Jaenes-Sánchez et al. [22] indicated that the acceptation of lockdown measures by the athletes was essential for the improvement of positive emotional states such as feelings of friendship.

Other factors that can affect mood states are age and experience. Regarding age, the study by Van de Groep et al. [23] pointed out that adolescents decreased their degree of empathy and increased social distance during lockdown. Regarding experience, the study by Mon-López et al. [13] emphasized that athletes with more experience suffered greater depression, while those with less competitive experience showed higher values of anxiety and stress.

The combination of lack of experience, age and psychological disorders produced by the lockdown may explain why the values of negative factors during week 1 of the present study were higher. As the weeks passed and de-escalation began in week 6, the negative mood states decreased their values in relation to the values of week 1, specifically in an extremely significant way for the depression factor. In the following phase of de-escalation (week 10), when athletes were able to train again with relative normality, the values of the negative factors decreased even more, showing significant differences in the depression factor, in the confusion factor and in the TMD in relation to the values obtained in week 1. At the same time, vigor, a positive factor, increased its value in relation to weeks 1 and 6. This trend towards improvement in mood behavior is shared by the study of Mehrsafar et al. [24], carried out in Iranian participants, and the study of Guicciardi and Pazzona [25], carried out in a sample of the Italian population. Furthermore, they point out that understanding the psychological implications of the restart phase can help to adopt appropriate behaviors for practicing sports in the COVID-19 situation.

The tension factor showed significant differences in the overall analysis, while the values obtained were lower than those provided by other research (10.21 ± 5.56 AU vs. 12.94 ± 6.228) [26]. The significant differences in the depression factor between week 1 and 10, when the values decreased, could be due to the positive effect of the athletes returning to training and being able to interact again with their coaches and teammates [27].

The anger values are higher than those found in other studies on elite athletes [28]. In fact, these values clearly modify the Iceberg profile, placing the anger percentile at the same level as that of the positive factor vigor, which shows higher values than those shown in other studies on athletes [27]. This increase in anger values could be explained by the lack of physical activity [9], as well as by the decrease in empathy and the increase in social distance in adolescent athletes [23].

When it comes to fatigue, the values obtained are lower than those obtained in other research on athletes. [13]. Moreover, these values do not show significant differences. Finally, by focusing on the confusion factor, we can point out that significant differences were obtained between the result values at the beginning and at the end of the study, and that these values are similar to those shown in other studies [27]. These differences observed between week 1 and week 10 may be due to the progress of the de-escalation process and the possibility of returning to training, as well as to the decrease in uncertainty caused by the restrictive measures imposed by the government in the first few weeks.

In Appendix A, we can see that there are significant differences between the various sports studied, especially in handball in relation to swimming and athletics. These significant differences suggest that athletes of team sports and individual sports use different strategies to confront the stressful situation derived from the pandemic. In this regard, the studies carried out point out that in individual sports, the coping strategy mainly used would focus on distancing from the competition, whereas in team sports, more task-oriented and emotion-oriented strategies are performed, due to the possibility of being supported by the team. [29,30].

When it comes to sex, in contrast to what the literature points out [12,13], no significant differences were found in our study in relation to any factor of the POMS or TMD. This situation could be plausible due to the fact that, being young, they lack the experience to face this type of situation (similar to one they would face in a major injury situation) [2]. Moreover, this is aggravated by the lack of “self-control” of these athletes [13]. Another factor that may justify the lack of significant differences is the number of subjects studied and the difference between the number of subjects of each sex.

One of the main limitations of this study is that, in regard to the pandemic, there are few articles that have used the same version of the POMS, which implies that comparisons must be made cautiously. On the other hand, the lack of previous knowledge of the mood states of our pre-COVID 19 population makes it difficult to compare results within the same population. In addition, other variables, such as mental health or the quality and quantity of sleep, could have been taken into account in order to acquire a more global vision of the situation suffered by our athletes.

## 5. Conclusions

The most prominent variations in the mood states profile of young athletes participating in high-level competitions during the different phases of lockdown were the significant decrease in the values of depression in weeks 6 and 10 with respect to week 1, of confusion in week 10 with respect to week 1, and of TMD in week 10 with respect to week 1. Furthermore, while comparing the various sports, significant differences were found between handball and athletics in the TMD at the three times of analysis, as well as in the values of depression, anger and confusion at different times at the end of lockdown and de-escalation. Finally, significant differences were found in the levels of tension and anger between handball and swimming. These conclusions could be taken into account when facing future situations with similar characteristics to optimize physical and mental performance.

## Figures and Tables

**Figure 1 ijerph-18-09326-f001:**
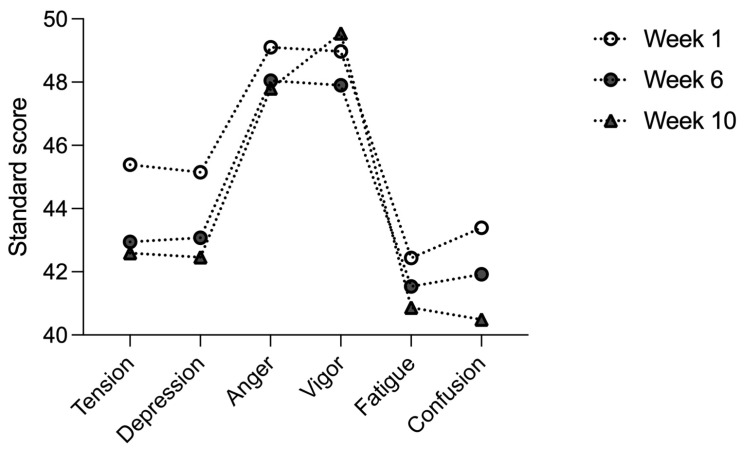
POMS profile of the 3 time points.

**Table 1 ijerph-18-09326-t001:** Sample characteristics.

Variable	Athletics	Handball	Swimming
*n*	18	16	11
Sex (M/F)	9/9	0/16	5/6
Age (years)	16.89 ± 2.05	15.94 ± 0.68	15.27 ± 1.10

Values of *n* and sex are presented as absolute frequencies. Values of age is presented as mean ± SD. M: male; F: female.

**Table 2 ijerph-18-09326-t002:** Summary of the mood state scores of the different time points.

Outcome	Week 1	Week 6	Week 10	*p* (Time)	*W*
TMD (AU)	126.55 ± 27.64	120.33 ± 26.60	117.65 ± 29.26 *	**0.001**	0.78
Tension (AU)	10.21 ± 5.56	8.44 ± 4.92	8.15 ± 5.82	0.070	0.76
Depression (AU)	8.79 ± 8.58	6.49 ± 6.88 *	5.85 ± 7.83 *	**<0.001**	0.70
Anger (AU)	8.97 ± 7.40	8.64 ± 7.11	8.08 ± 7.87	0.166	0.65
Vigor (AU)	14.95 ± 5.58	14.51 ± 5.26	15.23 ± 4.38	0.443	0.79
Fatigue (AU)	5.62 ± 4.33	5.08 ± 4.12	4.64 ± 4.59	0.245	0.73
Confusion (AU)	7.49 ± 4.67	6.72 ± 4.42	5.92 ± 5.00 *	**0.001**	0.75

Data are presented as mean ± SD. TMD: Total Mood Disturbance. AU: arbitrary units. W: Kendall’s W. Significant *p*-values (*p* ≤ 0.05) are shown in bold. * *p_Bonferroni_* ≤ 0.05 different to week 1 values.

## Data Availability

Data are available on request due to restrictions, i.e., privacy or ethical.

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
