# Peer review of "Variations in the Mood States during the Different Phases of COVID-19’s Lockdown in Young Athletes"

_ijerph, 2021, doi:10.3390/ijerph18179326_

Round 1
Reviewer 1 Report
I think the manuscript by Gavalda et al. for the most part is very well written. However, I believe the novelty for this type of research is low at the moment. At this point in time we understand what the pandemic has done to mental health and how lack of physical activity plays a role.
I believe the number of subjects for this study are too low and there are too few subjects per group to make comparisions between sports and sex.
You no males in your handball group making sport comparisons even more inappropriate as your results could be skewed by that fact.
Although not novel data this study would be publishable if there were a large sample size.
Reviewer 2 Report
The work is adequately written and clearly presented to the reader. The topic is quite interesting, dwelling on the impact of lockdown on the mood of athletes.
Despite these, there are some weaknesses
- The way in which the subjects in the sample were selected is not made clear
- The only instrument used is a Spanish one, which makes it difficult to replicate the study in populations of different ethnicity. Furthermore, this does not allow the results presented to be extended to different populations.
- Other variables that could have been useful for a better analysis of the differences between the groups of athletes involved are not analysed.
Reviewer 3 Report
Comments are in the attached file

Round 2
Reviewer 1 Report
The novelty of these data are low. Further your sample size is very small for this type of study. You have changed very little from your original version.
Author Response
Please see the attachment for the point-by-point response to the reviewer’s comments
